# Effects of High Glucose on Human Endothelial Cells Exposed to Simulated Microgravity

**DOI:** 10.3390/biom13020189

**Published:** 2023-01-17

**Authors:** Justina Jokšienė, Jayashree Sahana, Markus Wehland, Herbert Schulz, José Luis Cortés-Sánchez, Judit Prat-Duran, Daniela Grimm, Ulf Simonsen

**Affiliations:** 1Laboratory of Preclinical Drug Investigation, Institute of Cardiology, Lithuanian University of Health Sciences, Sukileliu Ave. 15, LT-50162 Kaunas, Lithuania; 2Department of Biomedicine, The Faculty of Health, Aarhus University, Ole Worms Allé 4, 8000 Aarhus, Denmark; 3Department of Microgravity and Translational Regenerative Medicine, Medical Faculty, Otto von Guericke University, Universitätsplatz 2, 39106 Magdeburg, Germany; 4Research Group “Magdeburger Arbeitsgemeinschaft für Forschung unter Raumfahrt- und Schwerelosigkeitsbedingungen” (MARS), Otto von Guericke University, 39106 Magdeburg, Germany

**Keywords:** microgravity, endothelial cells, hyperglycaemia, 3D cell culture, apoptosis, extracellular matrix, interleukin-8

## Abstract

A diabetogenic state induced by spaceflight provokes stress and health problems in astronauts. Microgravity (µ*g*) is one of the main stressors in space causing hyperglycaemia. However, the underlying molecular pathways and synergistic effects of µ*g* and hyperglycaemia are not fully understood. In this study, we investigated the effects of high glucose on EA.hy926 endothelial cells in simulated µ*g* (s-µ*g*) using a 3D clinostat and static normogravity (1*g*) conditions. After 14 days of cell culture under s-µ*g* and 1*g* conditions, we compared the expression of extracellular matrix (ECM), inflammation, glucose metabolism, and apoptosis-related genes and proteins through qPCR, immunofluorescence, and Western blot analyses, respectively. Apoptosis was evaluated via TUNEL staining. Gene interactions were examined via STRING analysis. Our results show that glucose concentrations had a weaker effect than altered gravity. µ*g* downregulated the ECM gene and protein expression and had a stronger influence on glucose metabolism than hyperglycaemia. Moreover, hyperglycaemia caused more pronounced changes in 3D cultures than in 2D cultures, including bigger and a greater number of spheroids, upregulation of NOX4 and the apoptotic proteins NF-κB and CASP3, and downregulation of fibronectin and transglutaminase-2. Our findings bring new insights into the possible molecular pathways involved in the diabetogenic vascular effects in µ*g*.

## 1. Introduction

Space travel, including the near-future expeditions to the Moon, Mars, and other planets and an increase in space tourism, sounds exciting and adventurous, but it possesses many health-related hazards. Stressors, such as noise, isolation, hypoxia, disrupted circadian rhythms, exposure to ionizing radiation, and microgravity (µ*g*), induce a cumulative and significant effect on health [1,2]. Prolonged exposure to the space environment causes alterations in glucose and lipid metabolism [3], and telomeric, epigenetic, ocular, and cognitive changes [4]. It has been suggested that, in particular, µ*g* causes subclinical diabetogenic changes. These manifest as insulin secretion and sensitivity alterations, decreased glucose tolerance and increased plasma glucose [5].

Glucose and insulin intolerance is a common complication during spaceflight [6] and its terrestrial s-µ*g* analogues, such as head-down tilt bed rest [7] or dry immersion [8]. Understanding how hyperglycaemic conditions in space affect astronauts’ health during long-term missions is crucial, and it is important to know and address all the subsequent health hazards during and after spaceflight. For instance, µ*g*, hyperglycaemia, and persistent low-grade inflammation are linked to chronic non-healing wounds. They are another relevant concern for space agencies [9]. Hyperglycaemia can trigger oxidative stress and inflammation via the AGE–RAGE axis by increasing TNFα, IL-6, and endothelin, and decreasing nitric oxide levels. Increased glucose availability generates more reactive oxygen species (ROS) and diminishes antioxidant surveillance, increasing oxidative stress. This, in turn, activates nonoxidative glucose pathways that further increase ROS production, stimulating the vicious metabolic cycle [10].

The exposure of endothelial cells to hyperglycaemia is linked with endothelial dysfunction and apoptosis [11]. High glucose also induces aerobic glycolysis and transglutaminase-2 (TG2) activity [12] and can trigger apoptosis via intracellular Ca^2+^ and ROS-induced activation of TG2 [13]. Another important fact is that endothelial cells are sensitive to mechanical forces as they are constantly influenced by shear stress, extracellular matrix (ECM) stiffness, mechanical stretch, and gravity. It was detected that endothelial cell apoptosis could be regulated via such mechanical forces [14]. However, the molecular mechanisms underlying the synergistic effects of µ*g* and hyperglycaemia are not fully understood. 

The principal aim of this study was to investigate the effects of hyperglycaemia on endothelial cells’ behaviour and key gene and protein expression in s-µ*g* using a 3D clinostat and static normogravity (1*g*) conditions. After 14 days of cell cultivation under s-µ*g* and 1*g* conditions, we harvested the cell material and measured the gene expression of ECM, inflammatory, glycolysis, glucose transport, and apoptosis-related genes. Apoptosis was further evaluated through TUNEL assay staining. Additionally, a Search Tool for the Retrieval of Interacting Genes/Proteins (STRING) analysis was performed to investigate the overall interactions among the genes and proteins of interest.

## 2. Materials and Methods

### 2.1. Cell Culturing

The immortalised cell line EA.hy926 (CRL-2922; ATCC, Manassas, VA, USA) was used in this study. It is a somatic cell hybrid type with endothelial cell morphology, displaying Weibel–Palade bodies that are specific to vascular endothelium, as previously described [15]. This cell line was derived by fusing primary human umbilical vein cells with a lung carcinoma A549 cell line. EA.hy926 cells (deposited by CS. Edgell) were purchased from ATCC (CRL-2922™).

The EA.hy926 cells were seeded into T25 flasks (Sarstedt, Nümbrecht, Germany) or slide flasks (Nunc™ Lab-Tek™ SlideFlask; Thermo Fisher Scientific, Waltham, MA, USA) for immunofluorescence staining and cultured using DMEM medium (D4947, Merck), supplemented with 10% FBS (F7524, Merck, Damstadt, Germany) and 1% penicillin/streptomycin solution (P4333, Merck, Damstadt, Germany), corresponding to low glucose (LG) medium. Half of the flasks were grown in hyperglycaemic conditions with high glucose (HG) medium, which was pre-made by supplementing regular 5 mM DMEM medium with D-(+)-Glucose solution (G8769, Merck, Damstadt, Germany). The cells were cultured either in 1*g* or s-µ*g* conditions with either LG or HG medium for 2 weeks. The medium was changed once after seven days. The experimental length was selected to recreate real µ*g* experiments in space, where good cell viability was demonstrated for at least 14 days [16]. 

### 2.2. Three-Dimensional Clinostat

Microgravity conditions were simulated using a custom-built 3D clinostat, which was designed and constructed by the German Space Agency (Deutsches Zentrum für Luft-und Raumfahrt, DLR, Cologne, Germany) and later revised by Aarhus University, Denmark [17]. 

The clinostats are the so-called National Aeronautics and Space Administration′s (NASA) and European Space Agency’s (ESA) acknowledged ground-based facilities and simulated µ*g* (s-µ*g*) on Earth [18]. These devices minimise the influence of gravity by rotating the biological sample around all three axes. 

To prepare the flasks for cultivation in s-µ*g* using the custom-built 3D clinostat, EA.hy926 cells were seeded at 5 × 10^5^ density into T25 flasks (n = 6 for each experimental condition) or at 3 × 10^5^ density into slide flasks (n = 6 for each experimental condition) in LG medium. Cell counting was performed using Corning^®^ Cell Counter (CLS6749, Merck, Darmstadt, Germany), and cell viability was assessed via staining with Trypan blue 0.4% solution (T8154, Sigma-Aldrich, Burlington, MA, USA). The cells were kept in 1*g* conditions (37 °C, 5% CO_2_) for 24 h to let the cells adhere. Then, serum starvation was implemented for another 24 h. Afterward, the FBS-free medium was removed, and the flasks were filled to the top with either LG or HG growth medium. The bubbles floating on top were removed through suction, and the lids were carefully screwed to avoid the formation of new air bubbles. In addition, the bottle caps were wrapped with sterilised parafilm and enclosed in sterilised sachets to prevent contamination. Then, half of the flasks were transferred from the laminar flow cabinet to the clinostat (37 °C) and mounted as close to the centre of rotation as possible. Another half of the flasks were used as static 1*g* controls and were placed in the incubator. The point of cell transfer to either clinorotation or incubator was considered the 1st day of the experiment.

The morphological evaluation of cell growth was performed using a light microscope (Leica DM IL LED, Germany) on the 1st, 7th, and 14th days of cell culture.

After seven days, half of the LG and HG medium from each flask was removed and replaced with complete fresh medium before allowing s-µ*g* flasks to stand vertically for 30 min to avoid sucking out the multicellular structures (MCS) and allowing them to settle down. After an additional week of cultivation in 1*g* and µ*g*, the cells, including adherent cells (AD) and MCS from s-µ*g* flasks, were collected. The spheroid count (n = 5) and spheroid area (n = 6) were calculated from the images obtained on day 14 with an EVOS M5000 microscope from s-µ*g* flasks (Thermo Scientific, Waltham, MA, USA).

### 2.3. Sample Collection and Protein Extraction

After 14 days, the cell samples for qPCR and Western blot experiments were collected. Media from 1*g* flasks were discarded, and media from s-µ*g* flasks were poured into 50 mL tubes to collect s-µ*g*-MCS. Each flask surface was washed three times with 5 mL of phosphate-buffered saline (PBS) and mechanically detached with cell scrapers (Sarstedt, Nümbrecht, Germany). The cell suspensions were collected in new 50 mL tubes (for 1*g* and s-µ*g*-AD cells). Then, all tubes containing the cells were centrifuged at 1000 rpm for 10 min at 4 °C. Supernatants (leaving approx. 1 mL) were removed, resuspended, and transferred to Eppendorf tubes. The Eppendorf tubes were centrifuged at 1000 rpm for 10 min at 4 °C, and the supernatant was fully removed. 

The pellets for qPCR experiments were loosened up in a small amount (100–200 µL) of RNAlater Stabilization Solution (Invitrogen by Thermo Fischer), resuspended in a total of 1 mL of RNAlater, transferred to Eppendorf tubes, and put into a −20 °C freezer until further use. 

The pellets for Western blot experiments were lysed with lysis buffer (RIPA with Halt protease and phosphatase inhibitor cocktail (78841, ThermoFisher Scientifics Waltham, MA, USA)) and protease/phosphatase inhibitor (1:100) by adding half the amount to pellet size, vortexed for 30 s and kept on ice for 30 min. During that time, the tubes were vortexed every 10 min for 30 s, followed by a sonification of 45 s using a Branson 2210 Ultrasonic bath (Buch & Holm, Herlev, Denmark). Then, the samples were centrifuged (10 min, 4 °C, 13,000 rcf), and the supernatants were transferred into new Eppendorf tubes. Protein concentration was determined using Lowry’s method.

### 2.4. Western Blot Analysis

The protein samples were boiled at 95 °C for 5 min. The wells of the Criterion TGX Stain-Free Gel (5678084, Bio-Rad, Hercules, CA, USA) were rinsed twice with 200 µL of the running buffer (1610772, Bio-Rad). Then, the samples were loaded, dedicating a few wells for a molecular weight marker (161-0373, Bio-Rad, Hercules, CA, USA). The gel ran for 30 min at 250 V with the running buffer. After the run, the gel was removed from the plastic and exposed to UV light to activate the gel and obtain an image for total protein quantification and normalization. During the gel UV scan, the polyvinylidene difluoride membrane (1620177, Bio-Rad, Hercules, CA, USA) was activated with ethanol 99.9% for 5 min, followed by washing the membrane with transfer buffer (10026938, Bio-Rad, Hercules, CA, USA). The filter papers were soaked in the transfer buffer for a few minutes before the transfer. The gel, the membranes, and the filter papers were stacked and put into the Trans-Blot Turbo Transfer System (Bio-Rad, Hercules, CA, USA) for 30 min at 100 V.

The membrane was read with UV light to show the total protein transferred and washed for 10 min with Tris-buffered saline-Tween 20 (TBST) to remove the excess transfer buffer. Then, the membrane was incubated for 5 min with the EveryBlot blocking buffer (12010020, Bio-Rad, Hercules, CA, USA). After blocking, the membrane was incubated with the primary antibody (Table A1) in a blocking buffer for 2 h at RT or overnight at 4 °C. Then the membrane was washed for 10–15 min 4 times with TBST, followed by incubation with the secondary antibody diluted in a blocking buffer for 2 h at RT. After appropriate washing steps, the membrane was developed using Clarity Western ECL Substrate (1705061, Bio-Rad, Hercules, CA, USA) solution according to the manufacturer’s instructions. Both reagents were mixed in a 1:1 ratio and spread on the membrane surface in the dark for 5 min. The images were obtained with Syngene PXi Touch (Syngene by Synoptics Group) and analyzed with the Syngene Gene Tools analysis software. Data are relative to total protein and expressed as % of control; n = 4 for NF-κB p65 and CASP-3 analysis, n = 5 for TG2, NOX4, and FN1. An overview of the used antibodies is given in Table A1.

### 2.5. Immunofluorescence Staining and Terminal Deoxynucleotidyl Transferase dUTP Nick End Labelling (TUNEL) Assay

Slide flasks after 14 days of cultivation were used for immunofluorescence staining. Media from 1*g* and s-µ*g* slide flasks were discarded. The cells were fixed with 4% PFA at room temperature (RT) for 20 min, then washed 3 times with PBS (3 min/wash). The permeabilization was performed with Triton-X 0.1% solution in PBS for 10 min, followed by a washing step. Then, the cells were blocked with 5% BSA in PBS for 30 min at RT. After the blocking step, the blocking solution was discarded, and the flask was detached from the slide with a special tool. 

The cell surface area was covered with corresponding primary antibody solutions and diluted in 1% BSA in PBS. The samples with the primary antibodies were kept in a humidified, light-protected box at 4 °C overnight. The next day, the cells were washed and incubated with the secondary antibodies for 1 h at RT in a humidified dark box. Then, the slides were rinsed with deionized water, preserved, and counterstained with Fluoroshield^TM^ with DAPI (F6057, Sigma-Aldrich, Burlington, MA, USA) mounting medium. The slides were sealed with cover glass and allowed to dry for 1–2 h before visualisation. The slides were stored at 4 °C protected from light. A list of antibodies and dilutions used is presented in Table A1. 

To evaluate the level of apoptosis, some slide flasks were analysed with Click-iT™ TUNEL Alexa Fluor Imaging Assay for Microscopy & HCS (C10245, Invitrogen by Thermo Fischer, Waltham, MA, USA), which was performed according to the manufacturer’s recommendations. This method was published in detail previously [19].

Immunofluorescence and TUNEL assay staining were analysed with an inverted confocal laser scanning microscope with super-resolution (LSM 800 Airyscan, Zeiss, Oberkochen, Germany). Images of TUNEL staining, triosephosphate isomerase 1, and fibronectin were obtained with a 40× air objective; all other images were acquired with a 40× oil immersion objective. Excitation and emission wavelengths were as follows: λ_exc_ = 495 nm and λ_em_ = 519 nm for FITC; λ_exc_ = 493 nm and λ_em_ = 517 nm for AF488; λ_exc_ = 557 nm and λ_em_ = 572 nm for AF546. All samples were analysed with the image analysis program ZEN 3.5 (Blue Edition). At least five fields of view were captured for TUNEL and each antibody staining (n = 5).

### 2.6. RNA Isolation and Quantitative Polymerase Chain Reaction (qPCR)

The RNeasy Mini Kit (Qiagen, Hilden, Germany) was used to isolate RNA according to the manufacturer’s instructions. RNA concentrations and quality were assessed spectrophotometrically at 260 nm using an Implen™ NanoPhotometer™ N60. The A260/280 ratio of isolated RNA was 1.5 or higher. The cDNA for quantitative real-time PCR (qPCR) was obtained using the High-Capacity cDNA Reverse Transcription Kit (Applied Biosystems, Darmstadt, Germany) using 1 µ*g* of total RNA in a 20-µL reaction mixture at RT. 

The qPCR method was used to determine the expression levels of selected genes after 14 d of incubation under s-µ*g* compared to the static control group (1*g*). The method was previously described in detail [20]. The primers were designed using NCBI Primer Blast, and they were selective for cDNA by spanning exon–exon junctions and had a Tm of around 60 °C. The primers were synthesised by TIB Molbiol (Berlin, Germany) and are listed in Table A2. All assays were run on an Applied Biosystems 7500 Fast Real-Time PCR System using the Fast SYBR™ Green PCR Master Mix (both Applied Biosystems, Darmstadt, Germany). 

For qPCR, there were 3 technical and 5 biological replicates. The reaction volume was 20 μL, including 1 μL of template cDNA and a final primer concentration of 500 nM. PCR conditions were as follows: 20 s at 95 °C, 40 cycles of 30 s at 95 °C and 30 s at 60 °C, followed by a melting curve analysis step (temperature gradient from 60 to 95 °C with +0.3 °C/cycle). All samples were measured in triplicate, and 18S rRNA was used as a housekeeping gene to normalise the expression data. The comparative Ct (ΔΔCt) method was used for the relative quantification of transcription levels, and the 1*g* control group was defined as 100% for reference.

### 2.7. STRING Analysis

To analyse the interaction network of genes used in qPCR, we used the STRING V11.5 tool [21] (available at https://string-db.org/, accessed on 5 December 2022) with a minimum interaction score of 0.4. The method was earlier published by Nassef et al. [19].

### 2.8. Statistical Analyses

All data are presented as mean ± standard deviation (SD). Statistical evaluation was performed using IBM SPSS Statistics 24 software (IBM, Armonk, NY, USA). Grubbs’ test was used to identify and remove statistical outliers. An unpaired two-tailed *t*-test was performed for analysis between 2 groups (spheroid count and spheroid area). The Mann–Whitney U Test was used to compare groups for qPCR and Western blot. Statistical significance was assumed when *p* < 0.05. 

## 3. Results

### 3.1. Hyperglycaemia in Microgravity Induces the Formation of Bigger and a Greater Number of Multicellular Structures

Images of the cells were taken on the 1st day before randomising them to the different gravity conditions, and on the 14th day after the cultivation in 1*g* and s-µ*g*. On day 1, in both LG and HG treatment groups, the cells were confluent and exhibited the same morphology as typically elongated endothelial cells (Figure 1a,d). After 14 days, distinctive differences were observed between the 1*g* and s-µ*g* groups, as the cells in the 1*g* were over-confluent and grew in layers on top of each other (Figure 1b,e), whereas the cells in the s-µ*g* samples formed MCS that floated freely in the supernatant and were detached from adherent cells (Figure 1c,f). In addition, s-µ*g* contained adherently growing cells on the bottom of the culture flasks. Tubular structures were seen in some of the s-µ*g* flasks, as shown in Figure 1f. 

Evaluation of the number and size of spheroids in LG and HG s-µ*g* samples revealed that there were significantly more spheroid structures in the HG group compared to the LG group. Additionally, the spheroid size was significantly increased in the HG s-µ*g* samples as opposed to the LG s-µ*g* samples (Figure 1g,h).

### 3.2. Extracellular Matrix Proteins Collagen IV and Fibronectin Are Downregulated in Microgravity

Collagen IV was more pronounced in 1*g* LG (Figure 2a) as opposed to 1*g* HG (Figure 1d); however, it seemed more expressed in s-µ*g* HG samples (Figure 2e,f) contrary to s-µ*g* LG (Figure 2b,c). Small collagen deposits had a punctate pattern in 1*g* HG (Figure 2d), whereas they showed a more fibrous pattern in s-µ*g*-AD HG and s-µ*g*-MCS HG samples (Figure 2e,f). s-µ*g*-AD and s-µ*g*-MCS samples cultured in hyperglycaemic conditions (Figure 2e,f) appeared to have a denser pattern and higher expression of collagen IV than the corresponding LG samples (Figure 2b,c). 

The relative gene expression for *COL4A6* was significantly downregulated in both the s-µ*g*-MCS LG and s-µ*g*-MCS HG groups compared to the corresponding 1*g* and s-µ*g*-AD groups (Figure 2g). There was no difference between LG and HG conditions regarding gene expression.

Fibronectin was sparse in the confocal microscopy images and was seen as small dots in the visual field in all LG samples (Figure 2h–j). In contrast, in the 1*g* HG sample, fibronectin was much more pronounced and had a fibrous mesh-forming appearance (Figure 2k). In the HG s-µ*g*-AD and s-µ*g*-MCS groups, the staining was reduced and had a pale and dusty appearance, more similar to the LG samples. 

The *FN1* gene expression was significantly decreased in s-µ*g*-MCS HG compared to 1*g* HG (Figure 2n). Western blot data showed a significant decrease in the protein expression of fibronectin in s-µ*g*-AD HG and s-µ*g*-MCS HG compared to the 1*g* HG control samples (Figure 2o). In contrast, only a significant depletion of FN1 protein expression in adherent cells was observed under LG conditions. Additionally, significantly lower FN1 protein expression was measured in s-µ*g*-HG groups compared to s-µ*g*-LG groups. Full blots are represented as Appendix A.

### 3.3. Microgravity Has a Stronger Effect on Glucose Metabolism Than Hyperglycaemia

#### 3.3.1. Glucose Transporters

GLUT1 (Figure 3a–f) and GLUT3 (Figure 3h–m) were present in all cells, and GLUT1 showed a pattern of perinuclear accumulation in all groups. GLUT3 seemed to have a similar pattern and intensity in all treatment groups. GLUT1 levels were slightly elevated in both s-µ*g*-LG and s-µ*g*-HG samples.

qPCR results of the *GLUT1* gene expression exhibited a significant downregulation of *GLUT1* between 1*g* control and s-µ*g*-MCS and between s-µ*g*-AD and s-µ*g*-MCS in both LG and HG. Irrespective of the glucose concentration, we observed a significant *GLUT3* increase between 1*g* control and s-µ*g*-AD and a considerable decrease in control levels between s-µ*g*-AD and s-µ*g*-MCS samples (Figure 3n).

#### 3.3.2. Triosephosphate Isomerase 1

The fluorescence intensity of triosephosphate isomerase 1 was higher in the s-µ*g*-AD LG and s-µ*g*-MCS LG samples compared to 1*g* LG (Figure 4a–c). It was also lower in the s-µ*g*-AD HG and s-µ*g*-MCS HG samples compared to 1*g* HG (Figure 4d–f). The *TPI1* gene expression was significantly downregulated in s-µ*g*-MCS LG samples compared to s-µ*g*-AD LG.

### 3.4. Transglutaminase-2 Expression Is Modulated by Microgravity and Hyperglycaemia

TG2 was present in all cells and showed a pattern of perinuclear accumulation in all groups (Figure 5a–f). Both LG and HG 1*g* samples exhibited a lower intensity and a much higher distribution in the corresponding s-µ*g*-AD and s-µ*g*-MCS samples. *TGM2* gene expression was significantly downregulated in HG s-µ*g*-MCS compared to both HG 1*g* control and HG s-µ*g*-AD cells (Figure 5g). In LG cells, only LG s-µ*g*-MCS vs. LG s-µ*g*-AD *TGM2* gene expression was significantly decreased. Relative protein levels were significantly elevated in LG s-µ*g*-MCS compared to the LG s-µ*g*-AD and LG 1*g* groups (Figure 5h). In addition, TG2 was more expressed in LG s-µ*g*-MCS than in the respective HG group (see also Appendix A).

### 3.5. NADPH Oxidase 4 and Interleukin-8 Are Upregulated on the Clinostat

#### 3.5.1. NADPH Oxidase 4

The NADPH oxidase 4 (NOX4) protein was detectable in all three groups, with increasing intensity from 1*g* control samples over s-µ*g*-AD to s-µ*g*-MCS cells, irrespective of glucose concentration (Figure 6a–f). Co-localisation with nuclei is observed due to two dyes overlapping and giving a greener colour around the nuclei area in s-µ*g*-AD and s-µ*g*-MCS samples. The s-µ*g*-AD groups showed a stronger fluorescence intensity compared with 1*g* samples. s-µ*g*-MCS exhibited overall the most intense and denser NOX4 deposits.

Relative gene expression of *NOX4* was downregulated in s-µ*g*-MCS, showing a statistically significant decrease in s-µ*g*-MCS HG compared to s-µ*g*-AD HG (Figure 6g). Increased protein expression was observed in HG s-µ*g*-AD, LG s-µ*g*-MCS, and HG s-µ*g*-MCS samples (Figure 6h). In addition, protein levels were significantly higher in HG compared to LG s-µ*g*-AD (see also Appendix A). 

#### 3.5.2. Interleukin-8

After the 14-day clinostat exposure, the interleukin-8 (CXCL-8) protein was detected either distributed in the cytoplasm (Figure 7, red arrows) or around the nucleus (Figure 7, yellow arrows). The signal intensities were lower in all LG samples compared to HG. Moreover, 1*g* LG and s-µ*g*-AD LG samples stored CXCL8 in dense deposits on cell edges, while corresponding HG groups had more intracellular granules distributed across the cytoplasm. Both LG and HG MCS showed higher CXCL8 fluorescence intensities. 

qPCR results for *CXCL-8* revealed a significant upregulation in both LG and HG s-µ*g* groups compared to the respective 1*g* samples (Figure 7g). The *CXCL8* gene expression in s-µ*g*-AD LG was approximately two-fold higher and more than four-fold higher in s-µ*g*-MCS LG compared to the corresponding 1*g* groups. In HG samples, the expression was almost four-fold higher in both clinorotated AD and MCS groups. 

### 3.6. Endothelial Cells Undergo Apoptosis on the Clinostat

#### 3.6.1. Osteopontin

Using confocal laser scanning microscopy (CLSM), osteopontin was visible in the cytoplasm of all samples (Figure 8a–f). The staining area had a lower fluorescence intensity in s-µ*g*-AD cells compared to s-µ*g*-MCS and 1*g*. 

Osteopontin is encoded by the *SPP1* gene. The qPCR results for *SPP1* (Secreted Phosphoprotein 1) revealed a significant upregulation between LG groups: a heightened elevation in clinorotated AD and MCS compared to 1*g*. The tendency for a similar increase was observed in HG samples. 

#### 3.6.2. NF-κB p65

NF-κB p65-positive cells were found in all cells. Staining for NF-κB p65 and visualising through CLSM revealed that in all adherent cells in both LG and HG groups, the protein was highly stored in the cytosol (Figure 9, red arrows). Some translocations of NF-κB p65 to the nucleus were detected in s-µ*g*-AD cells and a large number in the s-µ*g*-MCS group (Figure 9, yellow arrows). 

qPCR of the NF-κB p65 coding gene *RELA* showed significant downregulation of s-µ*g*-MCS HG compared with s-µ*g*-AD HG cells (Figure 9g). Western blot analysis revealed a significant and very high increase in all s-µ*g*-MCS samples compared to 1*g*. In addition, s-µ*g*-AD cells were influenced by the glucose concentration, which had an impact on the protein expression as a significant upregulation was observed in HG compared to LG (Figure 9h, Appendix A).

#### 3.6.3. Caspase-3

Caspase-3 positive cells were found in all samples (Figure 10, red arrows). Regarding the *CASP3* gene expression, significant up-regulations were detected in s-µ*g*-AD HG and s-µ*g*-MCS HG samples (Figure 10g). The Western blot analysis of cleaved CASP-3 revealed an upregulation in clinorotated AD HG compared to 1*g* HG and a downregulation to baseline levels in s-µ*g*-MCS HG compared to s-µ*g*-AD HG. A clear impact of hyperglycaemia was observed between s-µ*g*-AD samples, with a significant increase in the HG group (Figure 10h and Appendix A).

#### 3.6.4. TUNEL Staining

TUNEL assay and DAPI staining revealed a co-localisation of staining in the nucleus, which indicated the occurrence of apoptosis. As represented in Figure 11 (red arrows), a higher incidence of apoptotic cells was observed in cells cultivated on the clinostat.

## 4. Discussion

Endothelial cells are heterogenous mechanosensitive cells that undergo morphological, functional, and biochemical changes in µ*g*-conditions [22]. Exposure to real µ*g* (r-μ*g*) can only be achieved using parabolic flights, sounding rockets, space crafts, or space labs, as available on the International Space Station (ISS) [23]. However, high costs and the infrequency of missions limit the performance of such experiments [24]. In addition, a short duration of r-μ*g* in space missions limits the possibility of studies to investigate many lengthy and complex biological processes. Regarding parabolic flight experiments, rapid cycling between 1*g*, 1.8*g*, and μ*g* might also disrupt and interfere with the actual µ*g*-measurements [25]. Various methods to simulate µ*g* on Earth have been established to overcome these limitations, including 2D and 3D clinostats, random positioning machines, rotating wall vessels, and diamagnetic levitation [26]. However, only some aspects of r-μ*g* are mimicked [22]. In µ*g* analogues the extent of the Earth’s gravity vector cannot be removed; only its influence or effect can be reduced by randomising the direction of gravity over time (clinostat and random positioning machine) or by counteracting the gravitational force with another force (magnetic levitation). s-µ*g* analogues can only generate similar effects of μ*g* on physiological responses [27]. 

In the present study, we hypothesised that a high glucose concentration would affect the EA.hy926 cells cultured at 1*g* and µ*g* differently, including their behaviour and key gene and protein expression. Our data indicate that µ*g* had a stronger effect on gene and protein expression than hyperglycaemia. We observed a tendency for downregulation of the ECM genes *FN1* and *COL4A6* in s-µ*g*-MCS, irrespective of HG treatment. No change in *COL1A1* was observed (Figure A1). The fibronectin protein was also downregulated in s-µ*g* groups. Not so many studies have been conducted to investigate the ECM as a complementary counterpart of the gravireception system that consists of the cytoskeleton, ECM, and nucleoskeleton [28]. Our ECM findings differ from the previous similar study [29], though they correspond to the results obtained from r-μ*g* [30]. The variance may be due to different exposure times to µ*g* and the different µ*g*-simulating devices. 

Endothelial cells rely on glycolysis to produce energy for cellular metabolism [31]. Triosephosphate isomerase 1 (TPI1) is a key glycolytic enzyme that converts dihydroxyacetone phosphate to glyceraldehyde-3-phosphate [32]. In the study published by Bertelli et al. [33], a transcriptomic analysis revealed a downregulation of TPI1 after 4 weeks of culture in 25 mM D-glucose in comparison with controls. Similarly, the downregulation of glucose transporters is expected under the influence of HG treatment. Even though some studies did not indicate downregulations of glucose transporters in retinal capillary endothelial cells [34] or rat heart endothelial cells [35], vascular endothelial cells exposed to hyperglycaemia usually downregulate the rate of glucose transport by reducing *GLUT1* and *GLUT3* mRNAs and their protein expression. These regulations could be adaptive changes to protect endothelial cells from the damage caused by an excessive glucose influx [36,37]. Interestingly, our findings indicate no significant downregulation of the *GLUT1*, *GLUT3*, and *TPI1* gene expression due to hyperglycaemia. Downregulation of these genes was only caused by the change in gravity. Suppression of the glucose transport might have a connection with suppressed glucose metabolism in general, as was revealed in the study with EA.hy926 cells exposed to r-µ*g* on board the SJ-10 satellite, where the cells displayed a suppressed energy metabolism and glucose uptake [30].

Our study results show that the *TGM2* gene was downregulated and the TG2 protein was upregulated in µ*g*. There are no other studies that provide similar comparable data, but there is evidence that TG2, also known as tissue transglutaminase, is a cross-linking enzyme for fibronectin and mediates adhesion [38]. TG2 is a multifunctional enzyme, and its function in endothelial cell behaviour under µ*g* is unclear. It is known, for instance, that NF-κB activation depends on TG2 cross-linking of IκBα and subsequent proteasomal degradation, which suggests possible TG2 association with inflammation [39]. There is evidence that TG2’s function in endothelial cells depends on its location [40]. We observed a higher perinuclear intracellular TG2 accumulation in clinorotated samples, which might indicate its involvement in endothelial cell proliferation and apoptosis [40]. 

It has been shown previously that exposure to µ*g* upregulated NOX4 in some other cell types, such as blood mononuclear cells [41]. Some studies found that 4-week hindlimb unloading increased the levels of the pro-oxidative enzymes NOX2 and NOX4 in cerebral arteries but not in mesenteric arteries [42,43]. Our results indicate that NOX4 was significantly increased in s-µ*g*, which corresponds to other studies regarding µ*g* and ROS correlation [44]. The cytokine *CXCL8* was significantly elevated in s-µ*g*-AD and s-µ*g*-MCS, irrespective of HG treatment. These findings agree with earlier results from long-term s-µ*g* and r-µ*g* space missions with other cell types and seem to constitute a central reaction to µ*g*-exposure [45,46].

*RELA* encodes the ubiquitous transcription factor NF-κB and has a central position in the STRING protein–protein interaction profile of gene expressions studied using qPCR (Figure 9g and Figure 12b). The differential gene expression of *RELA* between adherent cells and spheroids under s-µ*g* in high glucose medium is accompanied by significant upregulation of the chemokine interleukin-8 which encodes the *CXCL8* gene. It should be noted that *CXCL8* regulation by s-µ*g* is independent of the medium composition and aggregation state of the cells (Figure 12b). Recently it could be shown that in non-small cell lung carcinoma cells but not in A549 adenocarcinoma human alveolar basal epithelial cells, the interleukin-8 secretion is influenced by TNF-related apoptosis-inducing ligand (TRAIL) receptors [47]. In A549 cells, *CXCL8* gene expression is regulated by JNK or MEK MAP kinases and ATF4, in addition to NF-κB [47]. In EA.hy926, interestingly, the mitogen-activated protein kinase 8 encoding *MAPK8* is also significantly upregulated in adherent cells compared to spheroids (Figure 12b and Figure A1), just like in NF-κB.

In general, the activation of programmed cell death was observed in this study. The TUNEL assay showed a higher incidence of apoptotic cells in s-µ*g* groups. Additionally, cleaved caspase-3 and the NF-κB protein were elevated in clinorotation groups. The NF-κB protein was also translocated to the nucleus in s-µ*g* samples; thus, the linkage between apoptosis and the pro-apoptotic NF-κB pathway is predicted. In addition, Kang et al. found that the s-µ*g* environment induced microvascular endothelial cell apoptosis and correlated with an increased expression of NF-κB [48]. The osteopontin levels were increased under s-µ*g*, similar to previous studies [49]. The overexpression of the cell adhesion molecule osteopontin is associated with cytoskeletal remodelling and the transition from 2D to 3D growth of endothelial cells under s-μ*g* [50,51]. It has been demonstrated previously that osteopontin has protective effects against apoptosis in endothelial cells [51]. No µ*g*-effects were observed in the regulation of other apoptotic genetic markers, including *CASP8*, *CASP9*, and *PARP1* (Figure A1). 

Based on our results, hyperglycaemia had no effect on gene and protein expression in normogravity conditions. In µ*g*-conditions resulting in detached cellular structures, including spheroids and multicellular structures, hyperglycaemia increased the size and the number of spheroid structures, decreased fibronectin and transglutaminase-2, and increased NF-κB, NOX4, and caspase-3. 

These findings are in accordance with other studies. It was found that spheroids from human dermal-derived microvascular endothelial cells of healthy donors were smaller and more compact, while spheroids from the cells of diabetic patients were larger in diameter and produced more sprouts [52]. Additionally, the activation of NF-κB signalling (NF-κB/miR-425-5p/MCT4 axis) showed a subsequent induction of apoptosis in endothelial cells under hyperglycaemic conditions [31], and caspase-3 was involved in high glucose-induced apoptosis, as demonstrated by Ho et al. [53]. On the other hand, contrary to our results, other studies found an increase in fibronectin and transglutaminase-2 in hyperglycaemic conditions in endothelial cells [13,54]. This suggests that µ*g* might interfere with hyperglycaemia-triggered pathways. 

Overall, our findings show that hyperglycaemia has marked effects in s-µ*g* but not in 1*g*. An overview of all other qPCR results not shown in the results section, including *COL1A1*, *CASP8*, *CASP9*, *PARP1*, and others, is given in Figure A1.

The current study had some limitations, including the evaluation at a fixed time point after 14 days. It is unclear how HG and µ*g* would gradually affect the cell behaviour and phenotype over time. In addition, the study was performed only for 14 days. Long-term effects lasting over 1–2 months in length might be required in future studies. Some of the HG effects may be lost due to the weaker sensitivity to the permanent EA.hy926 cell line stressor compared to primary cells [55]. To better understand the interactions of HG and µ*g*, it might be necessary to evaluate these factors in primary cells or in vivo and co-culture spheroids to make future findings more in vivo-relevant. Diabetogenic states induced in µ*g* might also be related to multisystemic effects. For instance, there might be a connection between skeletal muscle [56] and bone loss [5] or skeletal muscle glycogen synthesis [57]. Possible bone loss and hyperglycaemia connections in µ*g* might be due to increased glucocorticoids [58]. Increased skeletal muscle glycogen synthesis and overcompensation mean blood glucose depletion that might trigger a higher glucose intake or liver release. In addition, compromised liver carbohydrate metabolism might play a role [59]. These interconnections are not apparent and could be further examined in future studies. 

## 5. Conclusions

Overall, µ*g* revealed a stronger effect on the EA.hy926 cell line’s gene and protein expression as opposed to hyperglycaemia. Though, in some cases, a significant effect of hyperglycaemia in µ*g* was observed, such as the stimulation of a higher count and larger-sized MCS formation, the upregulation of the expression of the pro-oxidative enzyme NOX4 and the apoptotic proteins NF-κB and CASP3, and a more pronounced downregulation of fibronectin and transglutaminase-2. We observed an elevated expression of *NOX4* and *CXCL8* due to µ*g*, indicating increased oxidative stress and inflammation. Hyperglycaemia did not cause significant changes in glucose metabolism, and the changes occurred only due to the change in gravity. Our findings bring new insights into the diabetogenic vascular effects of µ*g* and their underlying mechanisms. 

## Figures and Tables

**Figure 1 biomolecules-13-00189-f001:**
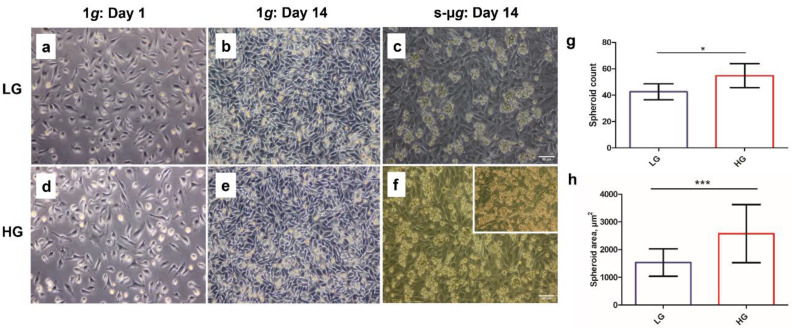
Morphological characteristics of EA.hy926 cells after exposure to low (LG) and high glucose (HG) media, and static normogravity (1*g*) or simulated microgravity (s-µ*g*) through phase contrast microscopy (**a**–**f**). Adherent cells (s-µ*g*-AD) and multicellular structures (s-µ*g*-MCS) in s-µ*g* samples can be observed (**c**,**f**). The insert in (**f**) shows another sample with a tubular structure. The scale bar represents 90 µm. The quantification of the spheroid number (**g**) and the relative spheroid size (**h**) are displayed. The spheroid size was determined via the spheroid area (µm^2^). Significant changes in spheroid number or spheroid size are indicated by * (*p* < 0.05) or *** (*p* < 0.001), respectively.

**Figure 2 biomolecules-13-00189-f002:**
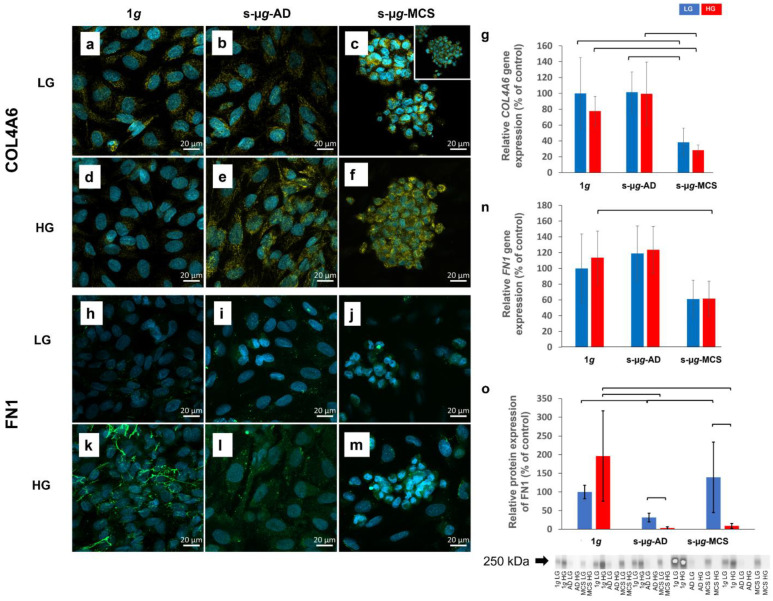
Analysis of the ECM proteins collagen IV (**a**–**g**) and fibronectin (**h**–**o**) after 14 days of cultivation either at 1*g* or s-µ*g* and either using low (LG) or high (HG) glucose medium. In the presentation of the s-µ*g* samples, a differentiation was made between AD and MCS. The data were detected via immunostaining (**a**–**f**,**h**–**m**), qPCR (**g**,**n**), and Western blot techniques (**o**). The brackets represent a significance level of *p* < 0.05.

**Figure 3 biomolecules-13-00189-f003:**
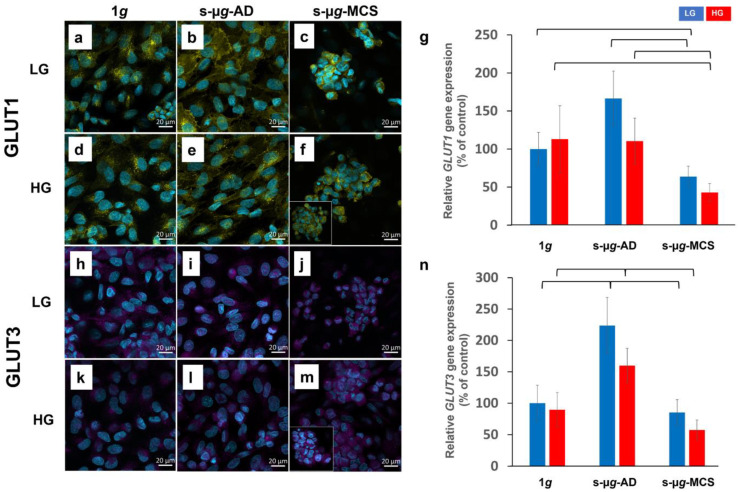
Analysis of glucose transporters GLUT1 and GLUT3 after 14 days of cultivation either at 1*g* or s-µ*g* and either using low (LG) or high (HG) glucose medium. Immunostaining (**a**–**f**,**h**–**m**) and qPCR (**g**,**n**) results are represented. The brackets represent a significance level of *p* < 0.05.

**Figure 4 biomolecules-13-00189-f004:**
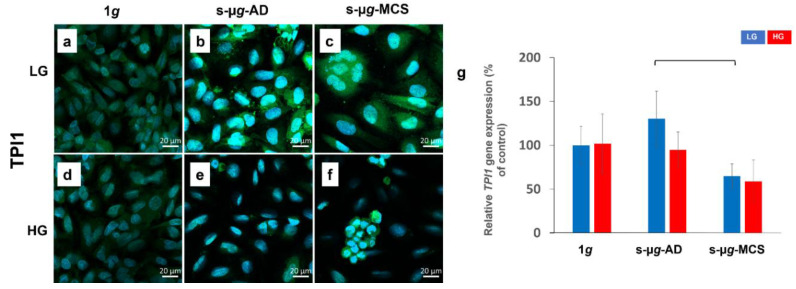
Analysis of TPI1 after 14 days of cultivation either at 1*g* or s-µ*g* and either using low (LG) or high (HG) glucose medium. Data were obtained through immunostaining (**a**–**f**) and qPCR (**g**). The bracket stands for a significance level of *p* < 0.05.

**Figure 5 biomolecules-13-00189-f005:**
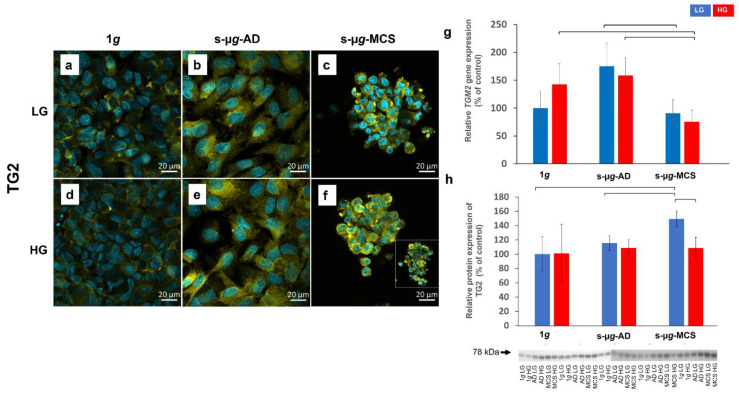
Analysis of TG2 after 14 days of cultivation either at 1*g* or s-µ*g* and either using low (LG) or high (HG) glucose medium. Data were obtained via immunostaining (**a**–**f**), qPCR (**g**), and Western blot (**h**) techniques. The brackets represent a significance level of *p* < 0.05.

**Figure 6 biomolecules-13-00189-f006:**
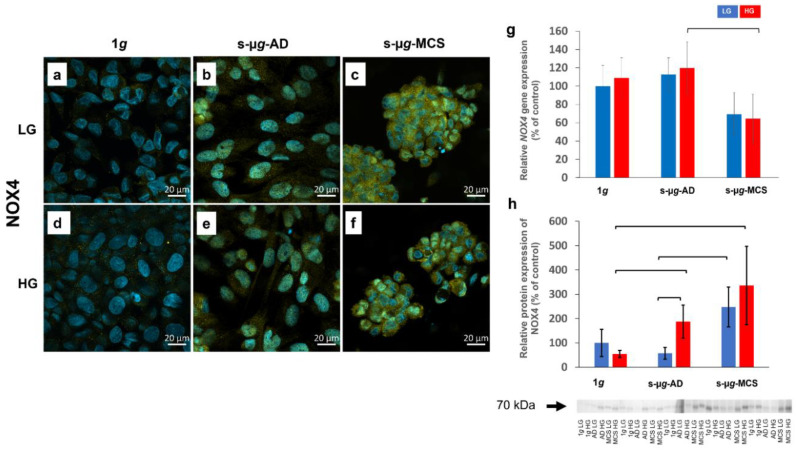
Analysis of NOX4 after 14 days of cultivation either at 1*g* or s-µ*g* and either using low (LG) or high (HG) glucose medium. Data were obtained via immunostaining (**a**–**f**), qPCR (**g**), and Western blot (**h**) techniques. The bracket stands for a significance level of *p* < 0.05.

**Figure 7 biomolecules-13-00189-f007:**
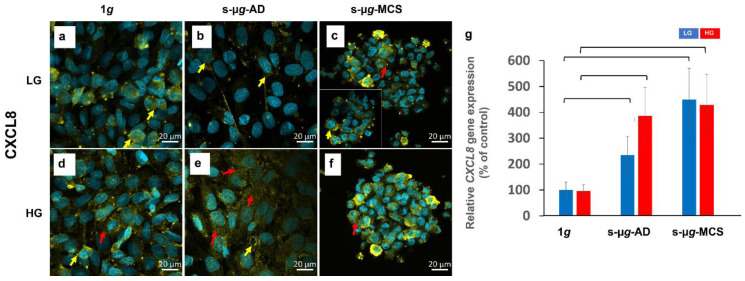
Analysis of CXCL8 after 14 days of cultivation either at 1*g* or s-µ*g* and either using low (LG) or high (HG) glucose medium. Data were obtained via immunostaining (**a**–**f**) and qPCR (**g**). The brackets stand for a significance level of *p* < 0.05.

**Figure 8 biomolecules-13-00189-f008:**
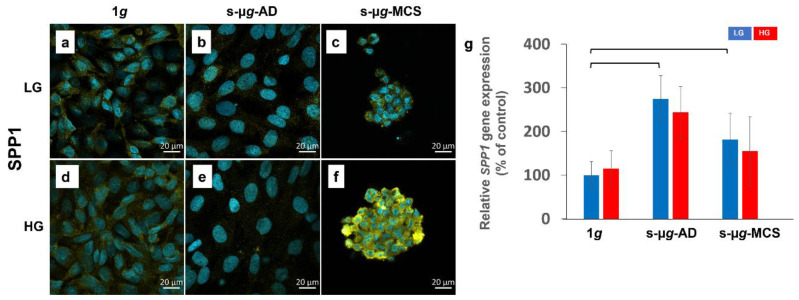
Analysis of SPP1 after 14 days of cultivation either at 1*g* or on the clinostat, either using low (LG) or high (HG) glucose medium. Data were obtained by immunostaining (**a**–**f**) and qPCR (**g**). The two brackets stand for a significance level of *p* < 0.05.

**Figure 9 biomolecules-13-00189-f009:**
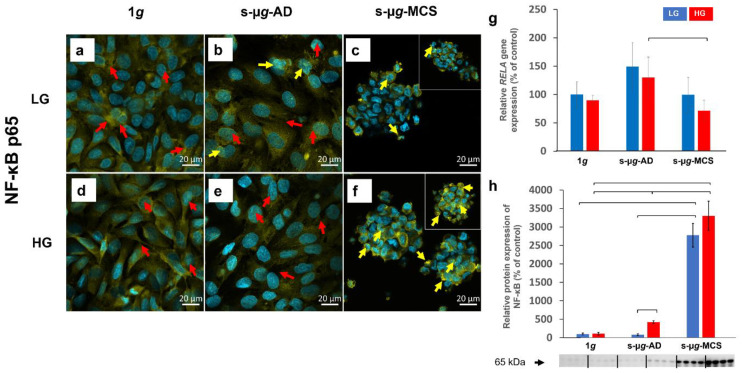
Analysis of NF-κB p65 after 14 days of cultivation either at 1*g* or on the clinostat, either using low (LG) or high (HG) glucose medium. Data were obtained via immunostaining (**a**–**f**), qPCR (**g**), and Western blot (**h**). The brackets stand for a significance level of *p* < 0.05.

**Figure 10 biomolecules-13-00189-f010:**
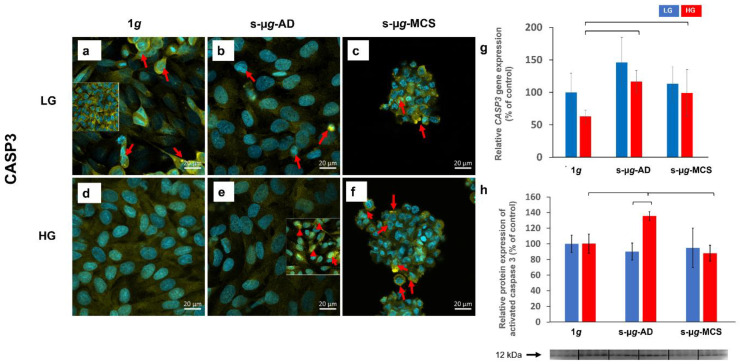
Analysis of CASP3 after 14 days of cultivation either at 1*g* or s-µ*g*, either using low (LG) or high (HG) glucose medium. Data were obtained via immunostaining (**a**–**f**), qPCR (**g**), and Western blot (**h**). The brackets represent a significance level of *p* < 0.05.

**Figure 11 biomolecules-13-00189-f011:**
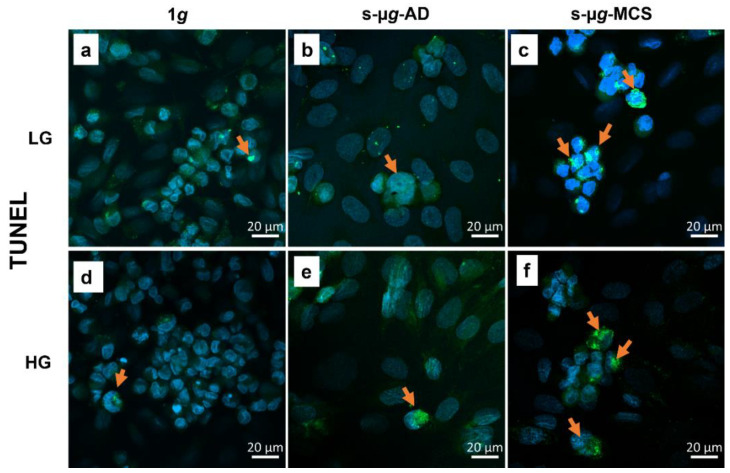
TUNEL assay after 14 days of cultivation, either at 1*g* or on the clinostat and either using low (LG) or high (HG) glucose medium. (**a**) 1*g* in LG; (**b**) s-µ*g* adherent cells in LG; (**c**) s-µ*g* multicellular structures in LG; (**d**) 1*g* in HG; (**e**) s-µ*g* adherent cells in HG; (**f**) s-µ*g* multicellular structures in HG.

**Figure 12 biomolecules-13-00189-f012:**
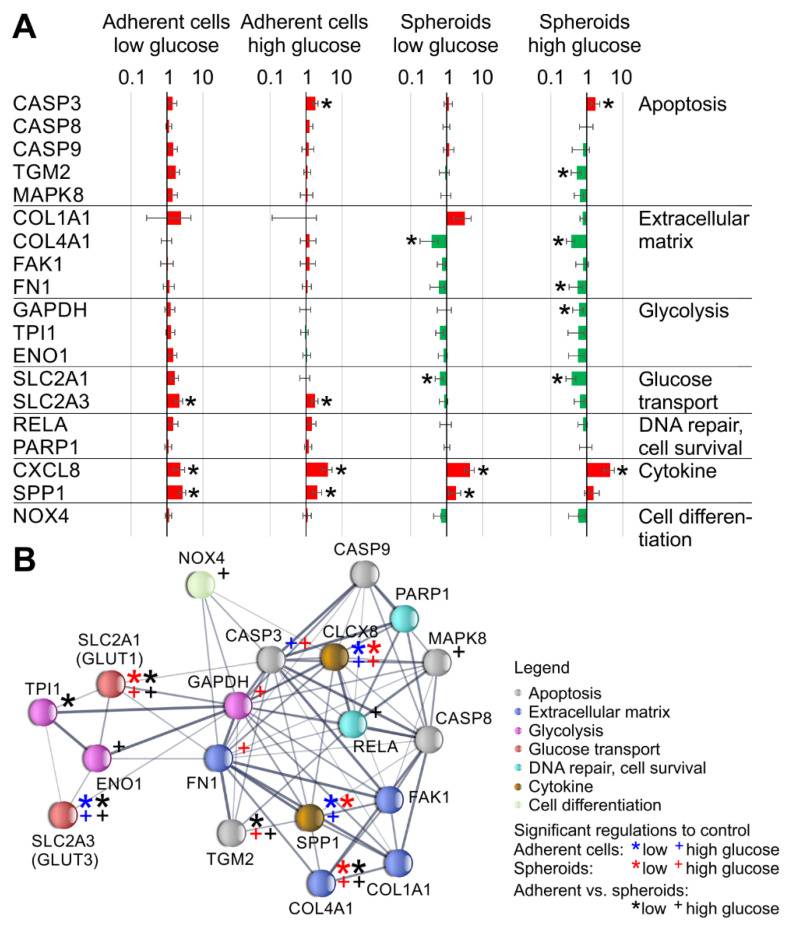
Overview of gene expression changes in the 19 genes quantified through qPCR. (**A**) Gene expression ratios. Significant regulations (*p* < 0.05) are indicated with an asterisk; upregulations or downregulations compared to the 1*g* control are indicated in red and green, respectively. (**B**) EMBL STRING protein–protein interaction of the 19 genes examined. Medium confidence interactions (0.4) with the colour-coded area of protein action are shown.

## Data Availability

Not applicable.

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
