# Peer review of "Effects of High Glucose on Human Endothelial Cells Exposed to Simulated Microgravity"

_biomolecules, 2023, doi:10.3390/biom13020189_

Round 1
Reviewer 1 Report
In their manuscript “Effects of high glucose on endothelial cells exposed to microgravity” Justina JokšienÄ— et al. address an interesting topic that is of relevance for future (long term) spaceflights as planned by ESA/ NASA. The authors describe the effects of hyperglycemia on EA.hy926 endothelial cells cultured under simulated microgravity conditions using a 3D clinostat. This device has been shown to simulate microgravity on a cellular level.
Simulated microgravity revealed a stronger effect on EA.hy926 cell line’s gene and protein ex- 565 pression as opposed to hyperglycaemia. As an effect of hyperglycemia under simulated microgravity, a higher formation of spheroids in number and size was observed. Up-regulation of apoptotic protein expression and more pronounced down-regulation of fibronectin and transglutaminase-2 were also detectable. Interestingly, hyperglycemia did not cause significant changes in glucose metabolism; the changes occurred only because of the change in gravity.
The research question is of high interest and clearly stated. The abstract is concise and informative. The structure of the article is clear. The material and methods, that were used, are appropriate and described clearly, the approach is innovative. The data and results are credible and feasible. All figures are understandable and of high quality. Limitations of the study are adequately discussed. References are comprehensive and up to date. In summary, this is a very good, well written manuscript that meets high standards.
Nevertheless, some minor changes seem to be necessary:
The title should refer to the simulated (!) microgravity.
You used a 3D clinostat for simulating microgravity on Earth. Are there other devices for creating ‘simulated microgravity’ available? Please explain this shortly in the discussion.
Please briefly mention why you chose an experiment period of 14 days.
The statistical tests are mentioned. However, there is no information about the number of experiments performed/cell culture flasks used for the respective tests.
Especially the images of the cells are of high quality. Please increase the font size of the y-axis labels for better readability.
Please standardize the spelling of NF-κB in the manuscript.
Author Response
In their manuscript “Effects of high glucose on endothelial cells exposed to microgravity” Justina JokšienÄ— et al. address an interesting topic that is of relevance for future (long term) spaceflights as planned by ESA/ NASA. The authors describe the effects of hyperglycemia on EA.hy926 endothelial cells cultured under simulated microgravity conditions using a 3D clinostat. This device has been shown to simulate microgravity on a cellular level.
Simulated microgravity revealed a stronger effect on EA.hy926 cell line’s gene and protein ex- 565 pression as opposed to hyperglycaemia. As an effect of hyperglycemia under simulated microgravity, a higher formation of spheroids in number and size was observed. Up-regulation of apoptotic protein expression and more pronounced down-regulation of fibronectin and transglutaminase-2 were also detectable. Interestingly, hyperglycemia did not cause significant changes in glucose metabolism; the changes occurred only because of the change in gravity.
The research question is of high interest and clearly stated. The abstract is concise and informative. The structure of the article is clear. The material and methods, that were used, are appropriate and described clearly, the approach is innovative. The data and results are credible and feasible. All figures are understandable and of high quality. Limitations of the study are adequately discussed. References are comprehensive and up to date. In summary, this is a very good, well written manuscript that meets high standards.
 We thank the reviewer for the positive comments and constructive response.
Nevertheless, some minor changes seem to be necessary: The title should refer to the simulated (!) microgravity.
 We agree, and the title now reads: “Effects of high glucose on human endothelial cells exposed to simulated microgravity”
You used a 3D clinostat for simulating microgravity on Earth. Are there other devices for creating ‘simulated microgravity’ available? Please explain this shortly in the discussion.
 Other methods for simulating microgravity on Earth are described on page 14, lines 455-462. We listed the examples of microgravity simulators on the same page, lines 455-457.
Please briefly mention why you chose an experiment period of 14 days.
 The experimental period of 14 days was chosen to simulate real microgravity experiments in space (Pietsch J et al., 2017). Please see page 2 lines 90-95.
The statistical tests are mentioned. However, there is no information about the number of experiments performed/cell culture flasks used for the respective tests.
 We performed 6 independent cell culture flasks for each experimental condition. The number of flasks and other experimental replicates is now stated in the methods section. Please see pages 3-5, lines 108-109, 129-130, 176-177, 208-209, 225.
Especially the images of the cells are of high quality. Please increase the font size of the y-axis labels for better readability.
 We agree, and the font size of the y-axis has been increased. Please see Figure 2-10, and Figure A1.
Please standardize the spelling of NF-κB in the manuscript.
 Corrected.
Author Response
The manuscript entitled “Effects of high glucose on endothelial cells exposed to microgravity” provides new information identifying gene expression changes that occur under conditions of microgravity and hyperglycemia in an endothelial cell line. Gene expression changes associated with the ECM, inflammation, glycolysis, and glucose transport were examined and the impact on protein expression was reported. Hyperglycaemia under microgravity conditions was shown to increase the size and number of spheroids, decreases fibronectin and transglutaminase-2 and increases NF-kB and caspase-3. This study provides new information defining the impact of microgravity along with high glucose on select endothelial cell transcripts.
 We thank the reviewer for the positive comments and constructive response.
Specific comments:
1. The data identifies gene expression changes in a single immortalized cell line. It would be beneficial to provide data from more than one endothelial cell line in order to generalize the conclusions obtained in this study. If available, the inclusion of data obtained in this model using primary endothelial cells in the experimental design would provide additional, useful insight, for this work.
 We agree with the reviewer that it would be indeed interesting to repeat these experiments with other primary endothelial cells. This, however, is not as straight forward as it may seem at first sight and would not be possible within a time frame of 10 days set for this revision. We have used primary endothelial cells exposed to simulated microgravity in the past (human saphenous vein endothelial cells (HSVECs) and human microvascular endothelial cells (HMVECs)) [1-3], however, all these experiments have been conducted on the RPM and not on the 3D clinostat and were considerably shorter with a maximum duration of 7 days, so viability tests over 14 days need to be conducted first. Furthermore, HSVECs did not form any spheroids and (admittedly shorter) studies on HUVECS by other groups also showed that these cells did not form 3D clusters in s-µg (table 1 in [3]). Lastly, it is generally difficult to generate enough primary cells to conduct these kinds of experiments, as they only have a small window of usable passages, which will mostly be consumed by the preparatory propagation of the cells. Studying, optimizing and eventually analyzing all these factors will require extensive preparatory work. Taking everything into consideration, we feel that, while addressing an important point, the amount and duration of the requested work would by far exceed this revision and is more suitable for a separate study. We have shortly addressed the issue in the discussion. Please see page 16 lines 556-560.
[1] Ma X, Sickmann A, Pietsch J, et al. Proteomic differences between microvascular endothelial cells and the EA.hy926 cell line forming three-dimensional structures. Proteomics. 2014;14(6):689-698. doi:10.1002/pmic.201300453
[2] Pietsch J, Gass S, Nebuloni S, et al. Three-dimensional growth of human endothelial cells in an automated cell culture experiment container during the SpaceX CRS-8 ISS space mission - The SPHEROIDS project. Biomaterials. 2017;124:126-156. doi:10.1016/j.biomaterials.2017.02.005
[3] Krüger M, Kopp S, Wehland M, et al. Growing blood vessels in space: Preparation studies of the SPHEROIDS project using related ground-based studies. Acta Astronaut 2019; 159:267-272. doi: 10.1016/j.actaastro.2019.03.074
2. The text indicates that the cells were confluent in both treatment groups on day 1. It is unclear why the cells were not initially sub-confluent when plated and samples compared over at least one timepoint prior to 14 days, rather than beginning with confluent cells and monitoring the change in gene expression at one timepoint two weeks later. Confluence alone will impact gene expression.
 The cells were seeded at the same density (Please see page 3 lines 107-110), kept overnight in an incubator to let cells adhere, and the FBS starvation for 24 hours was performed. Next day, either low or high glucose medium (with FBS) was added, and flasks were placed to clinostat. The day when flasks were placed into clinostat was considered experimental day 1, that is now shown in Figure 1. The experimental procedure is now clarified in the methods section, please see page 3 lines 107-122.
3. The cell counts after 14 days under low and high glucose conditions in simulated μg and 1g should be provided. The adherent cells in the 1g samples shown in the Figure 1 appear less confluent than the s-μg samples.
 We have now provided the initial cell counts in the text. Please see page 3 lines 108-109. Regarding cell counts after 14 days of clinorotation, it is no longer possible to count the cells in s-µg samples because of the formation of tubes and multicellular structures.
 Regarding Figure 1, we have now updated the pictures and the top labels, that were not correct and a bit confusing. 1g samples are more confluent after 14 days of cultivation compared to day 1, and s-μg samples did not need to be separated, as both adherent cells and MCS can be seen after 14 days in the visual field.
4. Based on the methods description, the cells were initially plated with FBS, then were kept for 24 hours without FBS followed by addition of medium with either of two concentrations of glucose. Please clarify – is it correct that FBS is added back to the media after 24 hours when the glucose is added and the 14 days commences? It is not clear why FBS was eliminated for 24 hours.
 We have now detailed in the methods that the cells were without FBS for 24h to enter same stage of cell cycle before being exposed to low and high glucose. Please see page 3 lines 113-115.
5. The legend to Figure 1 should define s-ug-AD and s-ug-MCS. The multicellular spheroids were defined as MCS in the text and it is clear that AD is the adherent population based on the text, but the abbreviation should be identified in the Figure legend.
 We have now updated Figure 1 legend text and defined the abbreviations. Please see legend to Figure 1.
6. GAPDH controls should be shown in the Figures that include Western blot data.
 The Western blot band intensities were normalized to the total protein in each lane. This was facilitated using the TGX stain-free gels, as described in the methods section (“After the run, the gel was removed from the plastic and exposed to UV light to obtain an image for total protein quantification and normalization”). GAPDH was not used as a housekeeping protein. Please see page 4 lines 159-161 and 178-179.
7. The legend to Figure A1 indicates that GAPDH was “mostly” not regulated during the 14day s-μg exposure except HG 1g control vs HG MCS. Were there other housekeeping genes used for this study in an effort to identify a housekeeping gene that is not impacted by the experimental conditions?
 As stated in the methods section, GAPDH was not used as a housekeeper for the qPCR analysis, but 18S rRNA, which was unregulated across all experimental groups. Please see page 5 lines 230-231.
Round 2
Reviewer 2 Report
The authors have adequately addressed all questions raised by this reviewer.